# Numerical Simulation and Multi-Objective Parameter Optimization of Inconel718 Coating Laser Cladding

**Sirui Yang** [1], **Haiqing Bai** [1,2,*], **Chaofan Li** [1], **Linsen Shu** [1,2], **Xinhe Zhang** [1] and **Zongqiang Jia** [1]

[1] School of Mechanical Engineering, Shaanxi University of Technology, Hanzhong 723000, China; ysr981124@163.com (S.Y.); lcf186394@126.com (C.L.); shulinsen19@163.com (L.S.); zxh030899@163.com (X.Z.); jzq5206@163.com (Z.J.)

[2] Shaanxi Key Laboratory of Industrial Automation, Hanzhong 723000, China

* Correspondence: bretmail@snut.edu.cn

**Abstract:** Aiming at the difficulty of temperature control in the laser cladding process of high-temperature nickel-based alloys, the influence of cladding parameters on the temperature of the molten pool, and the quality of the cladding layer were explored. Firstly, through the analysis of the finite element method, the Inconel718 single-pass cladding model was established on the surface of 45 steel by using parametric language and life–death element technology, the influence of different laser power and scanning speed on the temperature of the molten pool center was explored, and reasonable process parameters scope were selected. Secondly, taking the cladding parameters as independent variables, and taking the dilution rate and forming coefficient of the cladding layer as the response variables, using BBD (Box–Behnken Design) to design experiments the response surface analysis method was used to establish the regression prediction model of the cladding process parameters and response indicators, and the genetic algorithm was used to carry out multi-objective optimization to obtain the best results. The optimal parameter combination is a laser power of 1756 W, a scanning speed of 19.43 mm/s, and a powder feeding rate of 19.878 g/min. Finally, a multi-pass lap joint experiment was carried out with the optimal parameters, and it was found that the cladding layer has a dense and fine structure and a good metallurgical bond with the matrix, which can effectively guide the actual production.

**Keywords:** laser cladding; numerical simulation; temperature field; response surface method; multi-objective optimization

## 1. Introduction

Laser cladding is a dynamic instantaneous melting and solidification process, and the physical and chemical phenomena in the forming process are quite complex [1]. The temperature of the laser cladding process has a certain influence on the quality of the final cladding layer. However, the experimental conditions at this stage are not enough to monitor the temperature change of the molten pool in real-time. The finite element method is used to establish a numerical model that fits the reality. Simulation can be carried out to obtain the change of temperature field inside the molten pool, which provides a reliable means for process selection and optimization [2,3].

In the laser cladding process, there is an inevitable relationship between the temperature and the quality of the cladding layer, and there is a complex nonlinear relationship between the cladding parameters and the temperature of the molten pool [4]. It is of great significance to improve the quality of the cladding layer and provide theoretical guidance for the selection of laser cladding parameters [5,6]. Scholars have studied the temperature during the cladding process. By establishing a simulation model of laser cladding temperature and stress, it is found that the higher temperature changes are concentrated near the cladding layer, resulting in small-scale structural deformation in the area where the

cladding layer and the substrate are combined, and cracks are prone to occur. The effects of different cladding parameters on the temperature field were studied based on phenomena such as flow velocity, distribution, and fluid shape characteristics, and the validity of the established model was proved through experiments [7–10]. However, in order to choose appropriate process parameters, most scholars first use empirical methods to conduct a large number of experiments to determine the approximate range of process parameters and then use different mathematical methods to optimize [11–13]. These experimental materials would be wasted.

However, the research on the cladding layer of a high-temperature nickel-based alloy is mainly focused on the influence of temperature change on the cladding layer performance and structure, and there are few studies on how to control the temperature of the cladding process and select reasonable process parameters. This paper simulates the laser cladding process based on the finite element method and life–death element technology. First of all, the basic theoretical analysis and calculation of the temperature field establishes a single-channel laser cladding Inconel718 simulation model, sets reasonable model parameters, boundary conditions, constraints, etc., and obtains the temperature field distribution of the molten pool during the cladding process. It uses a reasonable range of process parameters, compares the experimental results with the simulation results, and verifies the feasibility of the model from the macroscopic size of the molten pool. Taking the cladding process parameters as independent variables and the cladding layer dilution rate and forming coefficients as the response variables, using the BBD (Box–Behnken Design) to design experiments, the response surface analysis method was used to establish a regression model of the response variables and indicators, and the genetic algorithm was used for multi-objective optimization to select the optimal one. The combination of parameters and multi-channel lap joint experiments showed that the metallurgical bonding was good and the surface had no obvious defects, which provided theoretical guidance for subsequent research.

## 2. Numerical Simulation

### 2.1. Theoretical Calculation

The laser cladding process is a dynamic physical metallurgical process with heat transfer, mass transfer, and instantaneous change. It is difficult for the current numerical simulation technology to fully consider all factors. Therefore, it is necessary to make reasonable simplifications and assumptions for the experimental process and consider the cladding process. The most influential factor in the process, reducing the complexity of the model, while increasing the time cost of simulation calculations. The relevant assumptions and simplifications are as follows:

(1)   Assuming that the cladding material is isotropic, the temperature is higher than the melting point, it is still processed in a solid state.
(2)   The material's specific heat, thermal conductivity, and other thermophysical parameters change with temperature, but the physical properties do not change with temperature [14].
(3)   The high-energy laser beam is assumed to be a moving heat source with a Gaussian distribution [15].
(4)   Assume that the initial temperature of the laser cladding environment is 20 degrees.

The temperature field of laser additive manufacturing is a typical nonlinear transient heat conduction problem, so the following heat conduction equation is used in this order value simulation:

$$\rho C(T)\frac{\partial(T\lambda(T))}{\partial t} = \frac{\partial}{\partial x}(\lambda\frac{\partial T}{\partial x}) + \frac{\partial}{\partial y}(\lambda\frac{\partial T}{\partial y}) + \frac{\partial}{\partial z}(\lambda\frac{\partial T}{\partial z}) + \alpha \tag{1}$$

where $C$ is the specific heat capacity of the material, $\rho$ is the material density, $\lambda$ is the thermal conductivity, $T$ is the temperature field distribution function, $\alpha$ is the internal heat source, $t$ is the heat transfer time, and $C$, $\lambda$ changes with time [16].

The Gaussian heat source model in which the heat flux density obeys the normal distribution satisfies the density expression equation:

$$Q = \frac{3\eta P}{\pi R^2} \exp\left(-\frac{3r^2}{R^2}\right) \tag{2}$$

where $Q$ is the heat flux density at the distance r from the center of the heat source, $\eta$ is the absorption rate of the laser energy by the workpiece, $R$ is the laser spot radius, $P$ is the laser power, and $r$ is the outer diameter of the Gaussian heat source distribution [17].

Before the laser beam and the powder reach the substrate, the energy exchange between the laser and the powder satisfies the following formula:

$$C_P \frac{dT_P}{dt} = eA_p(T_\infty - T_p) + A_p \varepsilon_p \sigma(\theta_R^4 - T_P^4) + \eta I \frac{A_P}{4} - m_p L_f \frac{df}{dt} \tag{3}$$

where $C_P$ is the specific heat capacity of the powder, $h$ is the heat exchange coefficient, $A_p$ is the surface area of the particles, $T_\infty$ is the temperature of the powder carrier gas, $T_p$ is the powder temperature, $\varepsilon_p$ is the radiation coefficient, $\sigma$ is the Boltzmann constant, $\theta_R$ is the radiation temperature, $\eta$ is the energy absorption efficiency of the powder particles, $I$ is the laser energy density, $m_p$ is powder quality, $L_f$ is the distance traveled by powder, and $f$ is coefficient of state of powder particles [18].

When setting the boundary condition of the laser cladding process, the initial temperature distribution and boundary conditions need to be given according to the specific environment of the experiment to facilitate the calculation and solution of the temperature field. There are three heat transfer forms in the laser additive remanufacturing process: heat conduction, convection, and radiation, so the boundary conditions are:

$$K_x(T) \frac{\partial(T)}{\partial(x)} n_x + K_y(T) \frac{\partial(T)}{\partial(y)} n_y + K_z(T) \frac{\partial(T)}{\partial(z)} n_z = \begin{cases} T_S(x, y, z, t) \\ q_s(x, y, z, t) \\ h(T_\alpha - T_S) \end{cases} \tag{4}$$

where $T_S$ is the boundary temperature, $q_s$ is the heat source and surface heat flux density, $T_\alpha$ is the ambient temperature of 20 degrees, $h$ is the convection coefficient, $n_x$, $n_y$, $n_z$ are the normals outside the boundary to obtain the direction cosine [19].

### 2.1.1. Materials and Properties

The base plate of this numerical simulation is 45 steel, and the cladding material is Inconel718 alloy. The basic physical parameters are shown in Table 1. The thermophysical properties of both vary with temperature, and their thermophysical parameters can be known by consulting the literature [20].

**Table 1.** Basic physical parameters.

| Material | Density (kg·m$^{-3}$) | Melting Point (°C) | Phase Transition Temperature Zone (°C) |
|---|---|---|---|
| 45 steel | 7850 | 1495 | 1490~1530 |
| Inconel718 | 8240 | 1260 | 1260~1320 |

### 2.1.2. Heat Source Loading and Model Setting

Based on the previous experimental results, to determine the size of the single-pass cladding layer, the base plate in the model is set to be $150 \times 50 \times 10$ mm$^3$, and the single cladding layer is $5 \times 5 \times 2$ mm$^3$. When the laser spot moves on the substrate in a certain direction at a constant speed, the powder is continuously fed to the substrate with the

movement of the laser, and the temperatures of the powder and the substrate rise rapidly and then drop rapidly, forming a cladding layer. The three-dimensional model established by the finite element software (ANSYS) is shown in Figure 1.

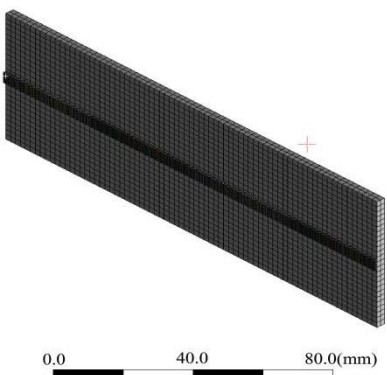

**Figure 1.** The 3D model.

The movement of the laser beam during the cladding process is simulated by means of cyclic load distribution. The cladding layer formation process is simulated by the "life and death cell" technique, that is, the laser beam moves to a certain unit block during the cladding process, and the unit block is activated, while the area where the laser beam does not pass, the unit block in this area is not activated [21], do not participate in the previous metallurgical bonding process of cladding materials, so as to realize the simulation of coaxial powder feeding laser cladding process.

### 2.2. Analysis of Laser Cladding Temperature Field

Numerical simulation of the temperature field distribution in the cladding process shows that the laser radiation center has the highest temperature, and the temperature gradually diffuses from the center to the outside. The gradient is denser than the rear, as expected for heat transfer. Through the previous experimental research, it is found that the laser power and scanning speed have the greatest influence on the quality of the cladding layer among the cladding parameters [22], so this simulation only considers the influence of these two parameters on the molten pool temperature.

#### 2.2.1. Temperature Field of Single-Pass Cladding under Different Laser Power

Figure 2 shows the cloud diagram of the temperature field of Inconel718 alloy cladding in a single pass with different laser powers. As the laser power changes from 1200 W to 2400 W, the maximum temperature of the molten pool changes from 1978 °C to 3698 °C. With the increase of laser power, the temperature range of the molten pool gradually increases, and the actual molten pool area also increases.

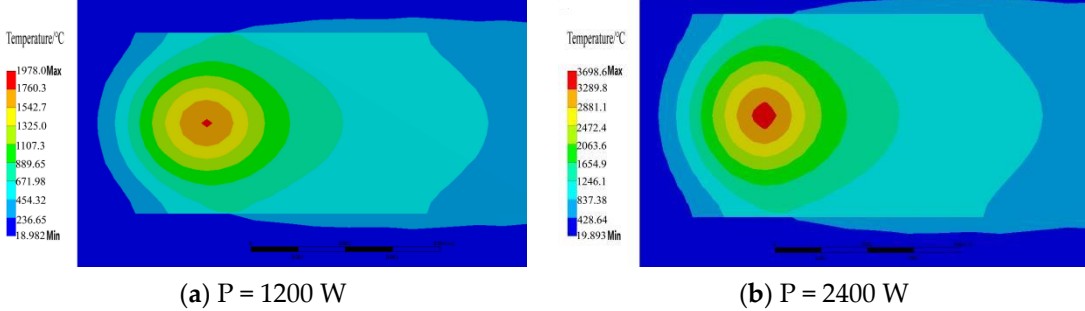

(**a**) P = 1200 W          (**b**) P = 2400 W

**Figure 2.** Cloud map of temperature field distribution under different laser power.

Therefore, the laser power range is selected as 1200 W–2400 W When the laser power is P = 1200 W (Figure 2a), the temperature exceeds the melting point of the cladding material powder, and a good cladding layer begins to form. When the power is lower than this, it is difficult for the cladding powder to form enough with the substrate metallurgical bonding. When the laser power is P = 2400 W (Figure 2b), the temperature in the molten pool is too high, causing the material to overburn, and the clad Inconel718 alloy may vaporize.

The temperature cycle curve of the molten pool at the same point under different laser power is shown in Figure 3. As the laser power increases from 1200 W to 2400 W, the peak value of the thermal cycle curve gradually increases, and each cycle curve has a similar heating–cooling process. The change in laser power affects the energy absorption of the substrate and powder materials. The higher the power, the more energy the material absorbs, resulting in a gradual increase in temperature.

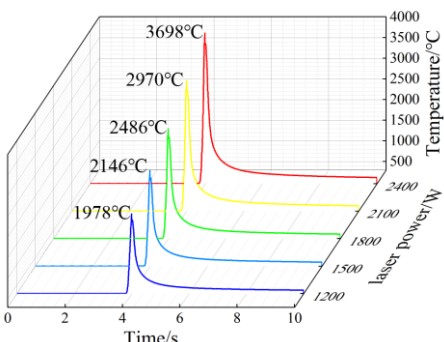

**Figure 3.** Variation of molten pool temperature under different laser powers at the same position.

Comparison of experimental and simulation results, the effect of laser power on the size of the molten pool is shown in Figure 4.

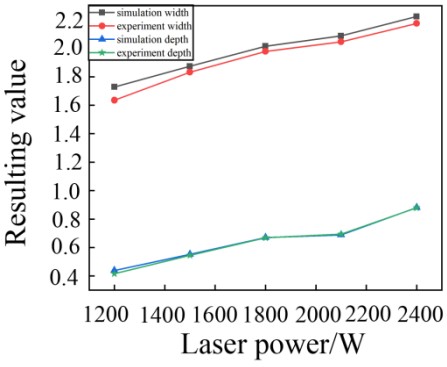

**Figure 4.** The variation rule of the laser power on the size of the molten pool.

When the laser power is low, the laser beam only melts a part of the powder in the spot to form a cladding track, making the molten pool area smaller. With the increase in laser power, the energy of the irradiation area of the substrate increases, so that the depth and width of the molten pool increase rapidly, and at the same time, more powder materials are melted to form a cladding layer, the height of the cladding layer also gradually increases. Therefore, the higher the laser power, the larger the molten pool area.

### 2.2.2. Temperature Field of Single-Pass Cladding at Different Scanning Speeds

The laser power is 1800 W, and the temperature field cloud diagram of laser cladding Inconel 718 alloy under different scanning speeds is shown in Figure 5. With the increase in scanning speed, the maximum temperature of the molten pool decreased from 2960 K to 2327 K. The temperature field has a similar change under different scanning speeds. The temperature distribution area in the center of the molten pool does not change significantly.

Compared with the laser power, the change in the scanning speed has no significant effect on the maximum temperature of the molten pool.

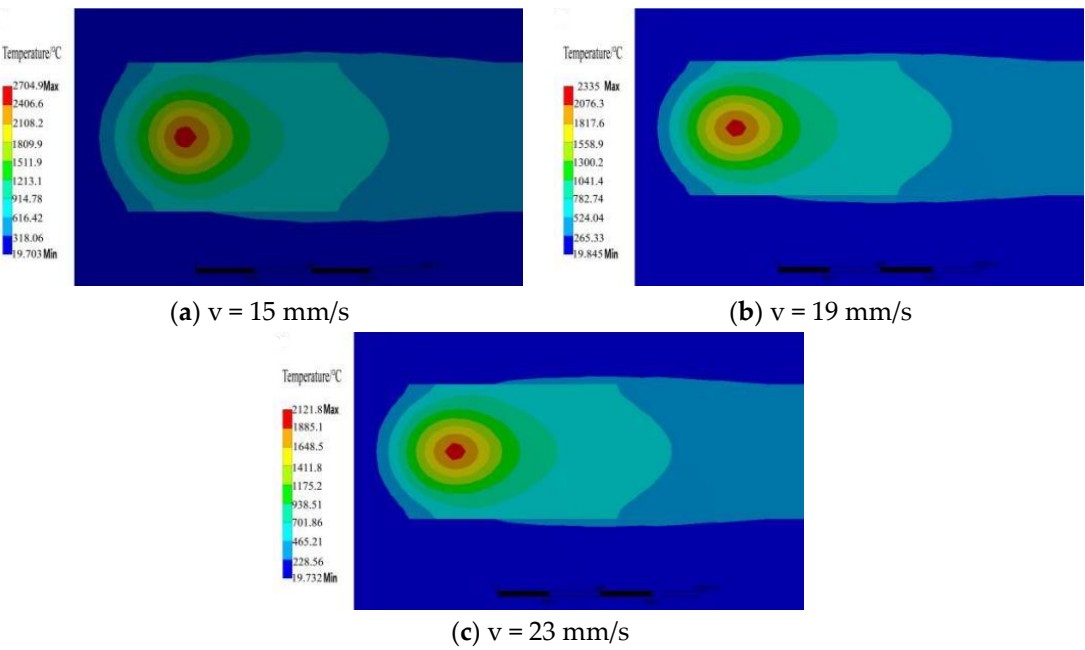

(**a**) v = 15 mm/s　　　　　　　　　　　　(**b**) v = 19 mm/s

(**c**) v = 23 mm/s

**Figure 5.** Cloud map of temperature field distribution under different scanning speeds.

Through numerical simulation, it is found that the scanning speed in the range of 15–23 mm/s can form a better molten pool morphology. Figure 6 shows the thermal cycle curves of the center of the molten pool at different scanning speeds. The thermal cycle curves have a similar change process, but the peak time is different. This is because the higher the scanning speed, the shorter the time to form the cladding layer of the same length. The scan speed just changes the energy absorbed by the substrate and powder per unit time.

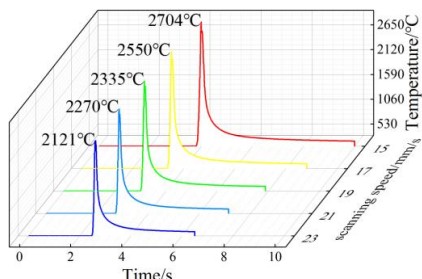

**Figure 6.** Variation of molten pool temperature under different scanning speeds.

Comparison of experimental and simulation results, the effect of scanning speed on the size of the molten pool is shown in Figure 7.

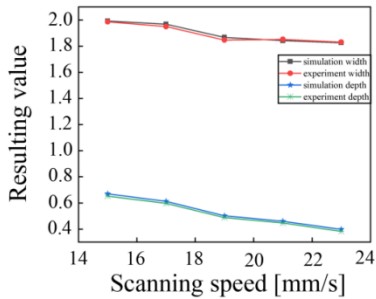

**Figure 7.** Variation rule of scanning speed on molten pool size.

The scanning speed affects the time that the laser heat source acts on the substrate. The lower the scanning speed, the more energy absorbed per unit area of the substrate and the larger the molten pool area. As the scanning speed increases, the energy absorbed by the substrate and the powder per unit area decreases, and most of the energy is lost, which reduces the area of the molten pool. Therefore, the higher the scanning speed, the smaller the molten pool area.

2.2.3. Numerical Simulation of Molten Pool Size and Experimental Verification

In order to verify whether the temperature field distribution of the established model is consistent with the actual situation, the size of the molten pool obtained by numerical simulation is compared with the experiment under the same parameters. In the central area of the molten pool, the temperature versus displacement distribution is plotted in the vertical scanning path width and depth direction, as shown in Figure 8, and the range above the melting point temperature is considered to be the width and depth of the molten pool [23].

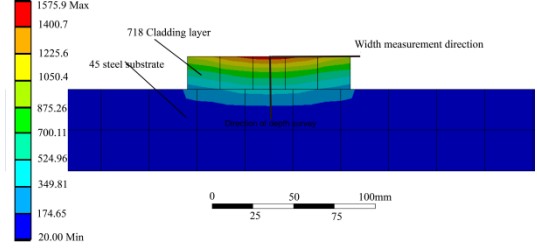

(**a**) Temperature measurement location and direction

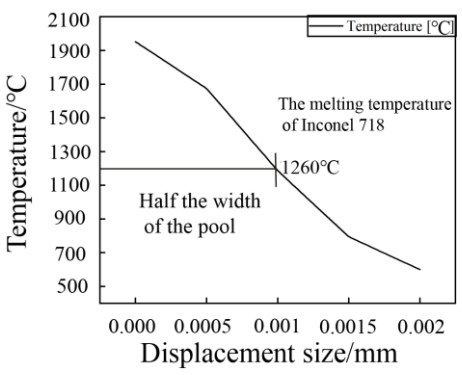

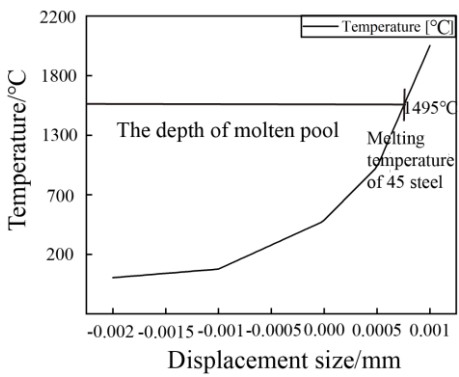

(**b**) Measurement of the width of the molten pool   (**c**) Determination of the depth of the molten pool

**Figure 8.** Schematic diagram of temperature measurement.

The size of the molten pool obtained by comparing experiments and simulations is shown in Table 2. The errors of the molten pool results obtained by experiment and simu-

lation are relatively small. It can be approximated that the established three-dimensional model, heat source function, and boundary condition settings are more reasonable.

**Table 2.** Comparison and error of molten pool size.

| Program | | Simulation Results (mm) | | Experimental Result (mm) | | Error | |
| --- | --- | --- | --- | --- | --- | --- | --- |
| Laser Power (W) | Scan Speed (mm/s) | Width | Depth | Width | Depth | Width (%) | Depth (%) |
| 1200 | 15 | 1.728 | 0.438 | 1.635 | 0.416 | 5.6 | 5.3 |
| 1800 | 19 | 1.867 | 0.502 | 1.847 | 0.488 | 1.1 | 2.8 |
| 1800 | 15 | 1.992 | 0.670 | 1.986 | 0.652 | 0.3 | 2.7 |
| 2400 | 15 | 2.223 | 0.881 | 2.175 | 0.879 | 2.2 | 0.2 |
| 1800 | 23 | 1.826 | 0.398 | 1.831 | 0.383 | 0.3 | 3.9 |

## 3. Laser Single-Pass Cladding Experiment

### 3.1. Experimental Equipment

The laser cladding experiment uses a 3 kW flexible fiber laser cladding system. The system consists of a YLS3000 semiconductor laser, ABB six-degree-of-freedom robot, ZF annular cladding head, CWFL water cooling device, and RH-DFOM double-tube powder feeding device (Figure 9). It mainly includes a software system, laser, numerical control system, powder conveying system, protective gas conveying device, and workbench. The 45 steel specimen with a substrate size of $150 \times 50 \times 10$ mm$^3$ was finely ground and polished before cladding to improve the flatness and wiped with anhydrous ethylene to remove the surface oil.

The sample observation equipment adopts an ultra-deep microscope (modelVHX-7000) (Figure 10), the section of Inconel 718 laser cladding is cut along the vertical direction of the cladding layer by wire cutting equipment, and then the section is ground and polished with sandpaper. Finally, the sample was corroded with 4% nitric acid alcohol for about 30 s, and the surface of the sample was wiped with alcohol and dried. The cross-section of the cladding layer was observed using a super-depth-of-field microscope. The cladding material is Inconel718 powder (Figure 11), and the powder is sealed and dried to ensure that it is dry, the cladding layer is Inconel718 alloy powder, the powder particle size is 40–106 μm, the bulk density is 4.1–4.6 g/cm$^3$, and the powder uniformity is high. The chemical compositions of the substrate and powder are shown in Table 3.

**Table 3.** Chemical composition of 45 steel and Fe45 powder (mass fraction, %).

| | C | Si | Mn | Ni | Cr | Fe |
| --- | --- | --- | --- | --- | --- | --- |
| 45 steel | 0.45 | 0.24 | 0.65 | 0.22 | 0.2 | margin |
| Inconel718 | 0.08 | 0.35 | 0.35 | 55 | 21 | margin |

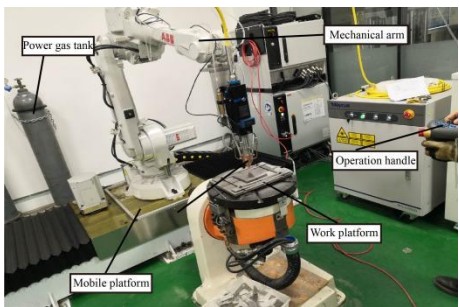

**Figure 9.** The 3 kW fiber laser cladding machine.

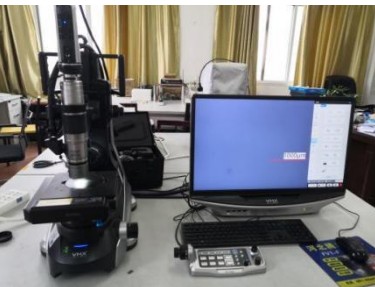

**Figure 10.** Super-depth-of-field microscope.

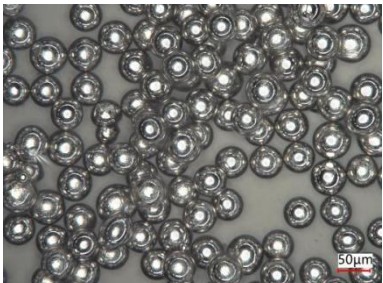

**Figure 11.** Microscopic image of Inconel718 alloy powder.

According to the previous experimental research, the distance between the nozzle and the surface of the worktable was adjusted to be 25 mm, the radius of the laser spot was 2 mm, the initial preheating temperature was set to 20 degrees, and argon (Ar) was used as the protective gas during the cladding process, shielding gas flow 6~10 L/min.

### 3.2. Experimental Design and Results

During the laser cladding process, the matrix and the cladding powder absorb different energy, resulting in different degrees of melting effect of the matrix and the powder. The partially melted matrix material will diffuse into the molten powder, which will have a certain dilution effect on the melted powder melt. We usually use the dilution rate to represent the diffusion result of the matrix material [24]. Within a certain range, the lower the dilution rate, the better, the calculation formula:

$$\eta = \frac{H}{h + H} \tag{5}$$

where $H$ is the depth of the molten pool, $h$ is the cladding height.

The overall quality of the cladding layer is represented by the forming coefficient of the cladding layer. Within a certain range, the larger the forming coefficient of the cladding layer, the better the forming quality of the cladding layer. The forming coefficient is the ratio of the width of the cladding layer to the depth of the molten pool. To express, the formula:

$$\varphi = \frac{W}{H} \tag{6}$$

where $W$ is the width of the cladding layer, $H$ is the depth of the molten pool.

Based on the response surface method, with laser power (A), scanning speed (B), and the powder feeding rate (C) as independent variables, and dilution rate as the response variable, BBD (Box–Behnken Design) was used to design the experiment. Within the scope of the process, use the response module of Design-Expert to design the experimental plan. The level codes of each factor determined by the numerical simulation results are shown in Table 4, and the specific experimental scheme and results are shown in Table 5.

**Table 4.** Cladding parameters and coding level table.

| Parameter Factor Level | −1 | 0 | 1 |
|---|---|---|---|
| Laser power (W) | 1200 | 1800 | 2400 |
| Scanning speed (mm/s) | 15 | 19 | 23 |
| Powder feeding rate (g/min) | 16 | 18 | 20 |

**Table 5.** Experimental scheme and results.

| | Laser Power (W) | Scanning Speed (mm/s) | Powder Feeding Rate (g/min) | H (mm) | H (mm) | W (mm) | $\eta$ | $\varphi$ |
|---|---|---|---|---|---|---|---|---|
| 1 | 1200 | 19 | 20 | 0.235 | 0.278 | 1.412 | 0.458 | 6.01 |
| 2 | 2400 | 15 | 18 | 0.763 | 0.458 | 2.087 | 0.625 | 2.73 |
| 3 | 1800 | 19 | 18 | 0.488 | 0.418 | 1.786 | 0.539 | 3.66 |
| 4 | 2400 | 23 | 18 | 0.623 | 0.475 | 1.925 | 0.567 | 3.09 |
| 5 | 1800 | 19 | 18 | 0.478 | 0.435 | 1.874 | 0.522 | 3.92 |
| 6 | 1800 | 23 | 20 | 0.335 | 0.325 | 1.554 | 0.507 | 4.64 |
| 7 | 2400 | 19 | 16 | 0.792 | 0.514 | 2.574 | 0.606 | 3.26 |
| 8 | 2400 | 19 | 20 | 0.554 | 0.448 | 1.882 | 0.553 | 3.40 |
| 9 | 1800 | 23 | 16 | 0.512 | 0.372 | 1.732 | 0.579 | 3.38 |
| 10 | 1200 | 23 | 18 | 0.356 | 0.330 | 1.381 | 0.519 | 3.88 |
| 11 | 1200 | 15 | 18 | 0.416 | 0.466 | 1.635 | 0.472 | 3.93 |
| 12 | 1800 | 15 | 20 | 0.445 | 0.354 | 1.758 | 0.557 | 3.95 |
| 13 | 1800 | 19 | 18 | 0.495 | 0.445 | 1.747 | 0.526 | 3.53 |
| 14 | 1800 | 19 | 18 | 0.503 | 0.441 | 1.755 | 0.531 | 3.49 |
| 15 | 1800 | 19 | 18 | 0.482 | 0.407 | 1.865 | 0.542 | 3.87 |
| 16 | 1800 | 15 | 16 | 0.665 | 0.432 | 2.134 | 0.606 | 3.21 |
| 17 | 1200 | 19 | 16 | 0.375 | 0.334 | 1.594 | 0.529 | 4.25 |

## 4. Results and Discussion

Based on the experimental results, the Design-Expert software was used to fit the above results, and a regression model of dilution rate, forming coefficient, and cladding parameters were established. The model was analyzed by variance analysis, as shown in Tables 6 and 7, where the P and F values were used to determine the significance of the analysis object, usually when the *p*-value is less than 0.05, the more significant the effect of this factor on the response value. The mean square indicates the degree of influence of the influence factor on the response value. The larger the mean square value of the influence factor, the more significant the influence of the factor on the response result. Lack of fit indicates the reliability of evaluating the fitted equation, and a value greater than 0.1 indicates that the model is significant and the fitted equation is good. $R^2$ represents the correlation of the model. The closer it is to 1, the better the correlation [25]. The correlation coefficient $R^2$ of the dilution rate model is 0.9708, and the correlation coefficient $R^2$ of the forming coefficient model is 0.9402. These values are all close to 1, so the model correlation is good. Adep Precision is a measure of the signal-to-noise ratio. A value greater than 4 can be used for simulation [26]. The signal-to-noise ratio for the dilution rate model is 19.723, and the signal-to-noise ratio for the forming factor model is 16.095, both of which are much larger than 4, so both models can be used for simulation.

It can be seen from Tables 5 and 6 that the laser power (A) and powder feeding rate (C) have very significant effects on the dilution rate and forming coefficient ($p < 0.0001$), the scanning speed (B), AB, and B2 have more significant effects on the dilution rate ($p < 0.05$), AC, $B^2$, $C^2$ had significant effects on the forming coefficient, and other factors had no significant effect on the dilution rate and forming coefficient. By comparing the mean square value, it is determined that the primary and secondary order affecting the dilution rate is A > C > AB > B > $B^2$. The primary and secondary order that affects the forming coefficient is A > C > $C^2$ > AC > $B^2$. After removing the factors that are not significant to the

response value, the quadratic regression fitting equation of different cladding parameters to the dilution rate and forming coefficient is finally obtained as follows:

$$\eta = 2.855 \times 10^{-4} \times A - 0.021 \times B - 0.013 \times C - 1.093 \times 10^{-5} \times AB \\ + 9.609 \times 10^{-4} \times B^2 + 0.67518 \tag{7}$$

where $\eta$ is dilution rate, $A$ is laser power, $B$ is scanning speed, and $C$ is powder feeding rate.

$$\varphi = 5.014 \times 10^{-3} \times A + 0.902 \times B - 4.425 \times C - 3.375 \times 10^{-4} \times AC \\ - 0.023 \times B^2 + 0.1457C^2 + 29.10998 \tag{8}$$

where $\varphi$ is forming coefficient, A is laser power, B is scanning speed, and C is powder feeding rate.

**Table 6.** Analysis of variance for dilution rate optimization model.

| Source | Sum of Square | Degree of Freedom | Mean Square | F | p | - |
|---|---|---|---|---|---|---|
| Model | 0.031 | 9 | 0.00346 | 25.85 | 0.0001 | significant |
| A | 0.017 | 1 | 0.017 | 130.12 | <0.0001 | - |
| B | 0.000968 | 1 | 0.000968 | 7.24 | 0.031 | - |
| C | 0.0075 | 1 | 0.0075 | 56.14 | 0.0001 | - |
| AB | 0.00276 | 1 | 0.00276 | 20.62 | 0.0027 | - |
| AC | 0.000081 | 1 | 0.000081 | 0.61 | 0.4618 | - |
| BC | 0.000132 | 1 | 0.000132 | 0.99 | 0.353 | - |
| $A^2$ | 0.000157 | 1 | 0.000157 | 1.17 | 0.3148 | - |
| $B^2$ | 0.00163 | 1 | 0.00163 | 12.16 | 0.0102 | - |
| $C^2$ | 0.000455 | 1 | 0.000455 | 3.41 | 0.1074 | - |
| Residual | 0.000936 | 7 | 0.000134 | - | - | - |
| Lack of Fit | 0.000651 | 3 | 0.000217 | 3.05 | 0.155 | not significant |
| Pure Error | 0.000285 | 4 | 0.0000712 | - | - | - |
| Cor Total | 0.032 | 16 | - | - | - | - |

**Table 7.** Variance analysis of forming factor optimization model.

| Source | Sum of Square | Degree of Freedom | Mean Square | F | p | - |
|---|---|---|---|---|---|---|
| Model | 8.15 | 9 | 0.91 | 12.23 | 0.0016 | significant |
| A | 3.92 | 1 | 3.92 | 52.91 | 0.0002 | - |
| B | 0.17 | 1 | 0.17 | 2.31 | 0.1724 | - |
| C | 1.91 | 1 | 1.91 | 25.79 | 0.0014 | - |
| AB | 0.042 | 1 | 0.042 | 0.57 | 0.4759 | - |
| AC | 0.65 | 1 | 0.65 | 8.75 | 0.0212 | - |
| BC | 0.068 | 1 | 0.068 | 0.91 | 0.3713 | - |
| $A^2$ | 0.022 | 1 | 0.022 | 0.3 | 0.5992 | - |
| $B^2$ | 0.54 | 1 | 0.54 | 7.35 | 0.0302 | - |
| $C^2$ | 0.89 | 1 | 0.89 | 12.05 | 0.0104 | - |
| Residual | 0.52 | 7 | 0.074 | - | - | - |
| Lack of Fit | 0.37 | 3 | 0.12 | 3.22 | 0.1439 | not significant |
| Pure Error | 0.15 | 4 | 0.038 | - | - | - |
| Cor Total | 8.67 | 16 | - | - | - | - |

Figure 12 shows the normal distribution of residuals of the model. The 17 groups of residuals are approximately linearly distributed on a straight line, indicating that the distribution of standardized residuals meets the requirements of normal distribution. The difference between the predicted and actual values of the equation is shown in Figure 13,

and all data points are evenly distributed on both sides of the line, further demonstrating the validity of the model.

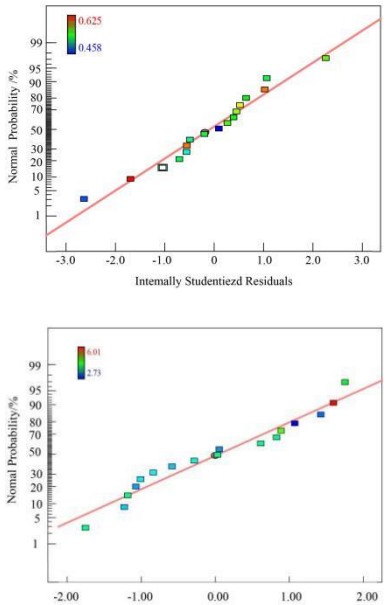

**Figure 12.** Normal distribution of residuals.

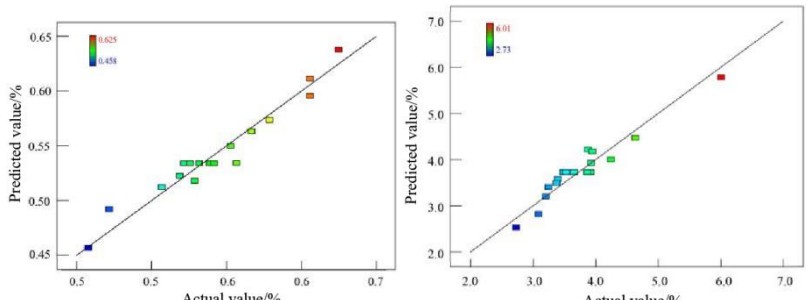

**Figure 13.** Relationship between the predicted value and actual value.

## 5. Multi-Objective Genetic Algorithm Optimization and Verification

The genetic algorithm is an adaptive probabilistic search algorithm that simulates the natural evolution process. It has better global optimization ability and is suitable for dealing with multi-objective optimization problems. The independent variables involved in multi-objective optimization are laser power $X_1$, scanning speed $X_2$, and powder feeding rate $X_3$. The objective function is the regression equation of dilution rate and forming coefficient. Taking the minimum dilution rate and maximum forming coefficient as the goal, a multi-objective optimization model and model constraints are established:

$$F_{\min} = [\eta, -\varphi]$$
$$s.t \begin{cases} 1200(\text{W}) \leq X_1 \leq 2400(\text{W}) \\ 15(\text{mm} \cdot \text{s}^{-1}) \leq X_2 \leq 23(\text{mm} \cdot \text{s}^{-1}) \\ 16(\text{g} \cdot \text{min}^{-1}) \leq X_3 \leq 20(\text{g} \cdot \text{min}^{-1}) \end{cases} \tag{9}$$

where $\eta$ is dilution rate, $\varphi$ is forming coefficient, $X_1$ is laser power, $X_2$ is scanning speed, and $X_3$ is powder feeding rate.

Solving using software writing programs based on the genetic algorithm. The population number was set to 100, the genetic algebra was 40, the crossover probability was 0.9, and the mutation probability was 0.01. Finally, the optimal solution was obtained as the

laser power of 1756 W, the scanning speed of 19.43 mm/s, and the powder feeding rate of 19.878 g/min.

Using the above conclusions, single-pass and multi-pass cladding experiments were carried out with a laser power of 1800 W, a scanning speed of 19 mm/s, a powder feeding rate of 20 g/min, and a lap rate of 50% [27]. The macroscopic morphology of the cladding layer was obtained. As shown in Figure 14.

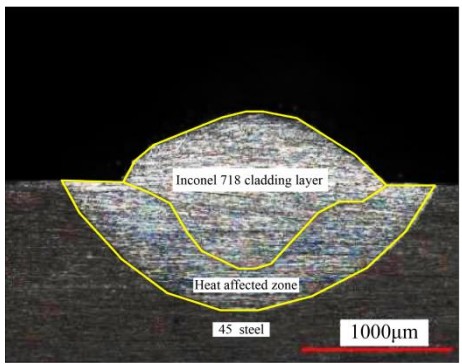
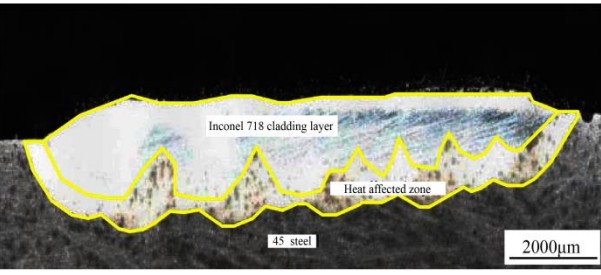

(**a**) Cross-sectional morphology of single-channel cladding.　　(**b**) Cross-sectional morphology of multi-channel cladding.

**Figure 14.** Surface morphology and cross-sectional morphology of cladding layer.

The surface morphology obtained by laser cladding under the optimal parameters is smooth and has no obvious defects, and the surface of the cladding layer with multiple overlaps is relatively flat, the heat-affected zone is narrow, and the metallurgical bond between the matrix and the cladding layer is good. The connection is tight, indicating that the quality of the cladding layer is high. Moreover, the measured macroscopic size of the molten pool is similar to the numerical simulation, indicating that the numerical simulation can effectively guide the actual production.

## 6. Conclusions

Based on the response surface analysis method, the laser power has the most significant impact on the quality of the cladding layer. The quadratic regression prediction model of the dilution rate and the forming coefficient was established, and the genetic algorithm was used for multi-objective optimization. The optimal parameter combination was obtained as the laser power of 1756 W, the scanning speed is 19.43 mm/s, and the powder feeding rate is 19.878 g/min. The verification experiments of single-pass and multi-pass cladding layers were carried out with the optimal parameters. The metallurgical bonding between the matrix and the cladding layer was good, indicating that the quality of the multi-layer cladding layer obtained under the optimal parameters was good.

**Author Contributions:** We declare that all authors contributed to the study conception and design. Material preparation, experimental data collection and analysis were performed by S.Y., C.L. and H.B. The first draft of the manuscript was written by S.Y.; S.Y., H.B., C.L., L.S., X.Z. and Z.J. commented on previous versions of the manuscript. All authors have read and agreed to the published version of the manuscript.

**Funding:** This research received no external funding.

**Institutional Review Board Statement:** Not applicable.

**Informed Consent Statement:** Not applicable.

**Data Availability Statement:** We declare that all data is true and available.

**Conflicts of Interest:** The authors declare no conflict of interest.

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
