# Peer review of "Numerical Simulation and Multi-Objective Parameter Optimization of Inconel718 Coating Laser Cladding"

_coatings, doi:10.3390/coatings12050708_

Round 1

Reviewer 1 Report

This paper simulated the laser cladding process based on the finite element method and life-death element technology, establishes a single-channel laser cladding Inconel718 simulation model, sets reasonable model parameters, boundary conditions, constraints, etc., and obtained the temperature field distribution of the molten pool during the cladding process.  Taking the cladding process parameters as independent variables and the cladding layer dilution rate and forming coefficient as
the response variables, the response surface analysis method was used to establish a regression model of the response variables and indicators, and the genetic algorithm was used for multi-objective optimization to select the optimal one. The verification experimentsofsingle-pass and multi-pass cladding layers werecarriedout with the optimal parameters. The metallurgical bonding between the matrix and the cladding layer was good, indicating the validity of the present approach.
  This paper is worth to be published becasuse many readers will be interested in the methodology to find the optimun experimental parameters.  However, subtle corrections are needed before the publication as follows:

p.2(left)  "Waste of resources." is not complete sentense.
p.2(right)  I think εp is not the powder temperature.  Give the exact explanation.
    The explanation of the last term in eq. (3) is missing.  Explain for m_p, L_f and f.

section 2.1.1 The author syas "their thermophysical parameters can be known by consulting the literature." but the concrete literature should be cited.

p.6(right)  The sentence "The lower the better" lacks subjects.

Fig.12  Check the label of x-axes.  Is "Studentiezd" correct?
Fig.13  Check the label of y-axes.  "Predictive" should be "Predicted".

The layout of figures and tables can be improved:
For Fig.2, the caption should be on the same page of the figure.
Table 4 sholud be printed in a single page. 
For Fig.13, the two figures  sholud be printed in the same page.

Author Response

Dear Reviewers:

Thank you for your letter and for the reviewers' comments concerning our manuscript entitled "Numerical simulation and multi-objective parameter optimization of Inconel718 coating laser cladding" (ID#coatings-1712737). Those comments are valuable and helpful for revising and improving our paper and the critical guiding significance to our research. We have studied comments carefully and made the correction which we hope to meet the approval. Revised portions are marked in red in the paper. Point-by-point responses to the reviewers’ comments are as follows:

Reviewers' comments and responses:

(1)The paper discusses numerical simulation and offers an optimal combination of parameters.

In the abstract it does not refer to the experimental part.

ResponseMany thanks for this suggestion.

Has been revised in the paper.In the abstract.

Added using BBD (Box-Behnken Design) to design experiments

(2)All the theoretical introduction is good but it is not applied to the study. Also in the introduction it does not make reference to the experimental part and how it is going to be related to the theoretical study.

ResponseMany thanks for this suggestion.

All the theoretical formulas are mainly for the necessary settings in the later simulation. These theoretical formulas are used in model establishment, material parameter setting, boundary condition setting, constraint condition setting, and heat source loading, heat conduction, energy transformation, etc. during temperature field simulation. To these theoretical formulas, but for the sake of the length of the paper, we will not repeat them in the paper.

An experimental part has been added to the introduction and a theoretical introduction part has been added.

(3)In equation (1) it says that , ρ、C and λare a function of temperature and temperature varies with time so , ρ、C and λ should also vary with time.

ResponseMany thanks for this suggestion.

Has been revised in the paper.In section 2.1

C and λ changes with time 

(4)In equations (1) and (2) the letter Q appears, but it means different things and can be misleading. The same happens with h in equations (3) and (4).

ResponseMany thanks for this suggestion.

Has been revised in the paper.In section 2.1

Q in equations (1) modified to αh in equations (3) has been modified to e.

(5)In point 2.1.2 it says "based on the previous experimental restls.." In which ones?

ResponseMany thanks for this suggestion.

Has been revised in the paper.In section 2.1.2

It is mainly the size of a single cladding layer, and its approximate range can be determined by experiment.

(6)The size of the figures of the legends and axes are very small.

ResponseMany thanks for this suggestion.

Has been revised in the paper.

The legends and axes in each figure have been modified.

(7)ARTICLEINFO insert space between“ARTICLE" and“INFO"

element technology, the influence  change "," by“." element technology. The influence..

ResponseMany thanks for this suggestion.

Has been revised in the paper.

A R T I C L E  I N F O

element technology. The influence..

(8)In (") you say that "physical properties do not change with temperature" but after equation (1) you say that density () change with temperature.

ResponseMany thanks for this suggestion.

Has been revised in the paper.In section 2.1

Writing errors, writing errors in equation (1),  do not vary with temperature.

(9)If λ and ρ depend on T then in equation (1) it should includeλ(T) and ρ(T) and as T varies with time then the equation would be

ResponseMany thanks for this suggestion.

Has been revised in the paper.In section 2.1

Writing errors, writing errors in equation (1),  ρ do not vary with temperature,But  λ varies with temperature,So equation (1) is modified as

(10)Table 1 Temperature in K.

Include a table with the composition of the materials 45 steel and Inconel718 alloy.

ResponseMany thanks for this suggestion.

Has been revised in the paper.

All temperature units in this paper are unified as degrees Celsius (°C).

Added the composition of the materials 45 steel and Inconel718 alloy.p.7(left),In section 3.1

the cladding layer is Inconel718 alloy powder, the powder particle size is 40-106µm, the bulk density is 4.1-4.6g/cm3, and the powder uniformity is high. The chemical compositions of the substrate and powder are shown in Table 3.

Table 3 Chemical composition of 45 steel and Fe45 powder

(mass fraction, %)

C

Si

Mn

Ni

Cr

Fe

45 steel

0.45

0.24

0.65

0.22

0.2

margin

Inconel718 

0.08

0.35

0.35

55

21

margin

(11)Why is important the shape and dimension of the base plate?

ResponseMany thanks for this suggestion.

In order to make the simulation and experiment closer, and make the substrate size of the simulation and experiment consistent, there is a certain relationship between the substrate size and the heat dissipation after laser scanning, so it is important to choose the appropriate substrate size.

(12)Increase the size of the micron bar of Fig.1

ResponseMany thanks for this suggestion.

The pictures in the paper have been modified.In section 2.1.2

(13)Figure 2a and figure 2b, to which laser energy does each image correspond? If is 1200W and 2400W, the maximum temperature 2234K and 3938K are different than in the images.

ResponseMany thanks for this suggestion.

Has been revised in the paper.In section 2.2.1

In the paper, Figure 2a corresponds to a laser power of 1200W, Figure 2b corresponds to a laser power of 2400W, and the maximum temperature of 2234K and 3938K corresponds to the maximum temperature in Figure 2, but the units are different, 1978℃=2234K, 3698℃=3938K, in order to ensure The unit in the figure is the same as that in the text, and the unit is now in degrees Celsius (°C).

(14)What is the accuracy of temperature measurements?, Degrees Celsius?, tenths of a degree?

ResponseMany thanks for this suggestion.

This article is Degrees Celsius

(15)In fig 2a and Fig2b you write 1325 (without decimals) or 18.982 (with 3 significant figures).

ResponseMany thanks for this suggestion.

Image 2a has been modified.In section 2.2.1

(16)"When the laser power is P=2400W (Fig.2 (b)), the temperature in the molten pool is too high and exceeds the boiling point of the alloy material, the cladding Inconel718 alloy may be vaporized" What is the boiling point? This data do not appear in table 1.

ResponseMany thanks for this suggestion.

Has been revised in the paper.In section 2.2.1

In the process of laser cladding, when the laser power is too high, the material will vaporize. In this paper, the boiling point is used to represent this phenomenon, which may not be accurate. After careful consideration, it is now chosen to use overburning to represent this phenomenon.

Therefore, the text should be changed to:When the laser power is P=2400W (Fig. 2(b)), the temperature in the molten pool is too high, causing the material to overburn, and the clad Inconel718 alloy may vaporize.

(17)It is impossible to read data in Fig.3

ResponseMany thanks for this suggestion.

Figure 3 mainly illustrates the temperature change of the molten pool at the same point when different laser powers change. The change process is similar, but the maximum temperature that can be reached at different laser powers is different.

Figure 3 has been modified.In section 2.2.1

(18)Fig.4 show different between simulations and experiment results. But in the text you do not talk about experimental test.

ResponseMany thanks for this suggestion.

Has been revised in the paper.In section 2.2.1

Comparison of experimental and simulation results,the effect of laser power on the size of the molten pool is shown in Fig.4.

(19)Fig.5 Velocity is written with a lowercase V.

ResponseMany thanks for this suggestion.

Has been revised in the paper.In section 2.2.2

Uppercase V has been modified to lowercase v.

(20)Fig 5 you show the range of speed that can form a better molten pool morphology. Can you show anyone that form a poor molten pool?

ResponseMany thanks for this suggestion.

The main purpose of this paper is to select the appropriate parameter range through the change of the molten pool morphology. When the scanning speed is too fast, the powder and the substrate will not be completely melted. When the scanning speed is too low, the powder and material will be overburned. , so it is only necessary to find a suitable molten pool morphology to represent the approximate range of parameters.

(21)It is impossible to read data in Fig.6

ResponseMany thanks for this suggestion.

Figure 6 has been modified.In section 2.2.2

(22)Fig.7 show different between simulations and experiment results. But in the text you do not talk about experimental test.

ResponseMany thanks for this suggestion.

Has been revised in the paper.In section 2.2.2

Comparison of experimental and simulation results,the effect of scanning speed on the size of the molten pool is shown in Fig.7.

(23)Table 2 Laser powder (W), scan speed (mm/s), width (mm), depth (mm)

ResponseMany thanks for this suggestion.

Has been revised in the paper.In section 2.2.3

Table 2 has been modified toLaser powder (W), scan speed (mm/s), width (mm), depth (mm)

(24)Table 2 How many measurements have you made to determine the error in %?

ResponseMany thanks for this suggestion.

Each parameter test was carried out three times, and the mean value was calculated. The mean value is compared with the simulated value for error comparison.

(25)Why the substrate size is 100mmx100mmx10mm and in the simulation the base plate (150x50x10 mm) is different?

ResponseMany thanks for this suggestion.

Has been revised in the paper.In section 3.1

Writing error, the size of the substrate should be (150x50x10 mm).

(26)You have write in the same way with mm at the end or with mm after each figure.

ResponseMany thanks for this suggestion.

Has been revised in the paper.

with mm at the end.

(27)After Fig.2 you write "Therefore, the laser power range is selected as 1200W-2400W"

ResponseMany thanks for this suggestion.

Has been revised in the paper.In section 2.2.1

Therefore, the laser power range is selected as 1200W-2400W.When the laser power is P=1200W (Fig. 2(a)), the temperature exceeds the melting point of the cladding material powder, and a good cladding layer begins to form. When the power is lower than this, it is difficult for the cladding powder to form enough with the substrate metallurgical bonding. When the laser power is P=2400W (Fig.2 (b)), the temperature in the molten pool is too high and exceeds the boiling point(about 3000℃) of the alloy material, the cladding Inconel718 alloy may be vaporized.

(28)Fig.9 It is impossible to read the words inside the image.

ResponseMany thanks for this suggestion.

Figure 9 has been modified.In section 3.1

(29)Fig.9 You have to write 9. 3kW instead of 9.3KW

ResponseMany thanks for this suggestion.

Modified to 3kW. In section 3.1

(30)It is necessary more data about the 3kW fiber laser cladding machine, powder feed, super depth of field microscope, Inconel718 alloy powder.. (manufacture, brand, model, ..

ResponseMany thanks for this suggestion.

Has been revised in the paper.In section 3.1

Added The sample observation equipment adopts an ultra-deep microscope(modelVHX-7000); The laser cladding experiment uses a 3kW flexible fiber laser cladding system. The system consists of YLS3000 semiconductor laser, ABB six-degree-of-freedom robot, ZF annular cladding head, CWFL water cooling device and RH-DFOM double-tube powder feeding device;Powder data for Inconel718 has been increased, the powder particle size is 40-106µm, the bulk density is 4.1-4.6g/cm3, and the powder uniformity is high.

(31)Table 3 Laser power (W), Scanning speed (mm/s), powder feeding rate (g/min)

ResponseMany thanks for this suggestion.

Has been revised in the paper.In section 3.2

Modified in Table 3 to Laser power (W), Scanning speed (mm/s), powder feeding rate (g/min)。

(32)Table 4. If you write the table in this way:

Why weren't all possible combinations made?

Why were some combinations of parameters repeated and others not?

ResponseMany thanks for this suggestion.

In order to reduce the number of experiments and reduce the consumption of materials, the BBD experimental design scheme in the response surface method can achieve the same results as the full factor experimental scheme, so it is not necessary to design all possible combinations of experimental schemes. The experimental design scheme is designed by Design-Expert software. This experimental design scheme can evaluate a non-linear relationship between indicators and factors, an experimental design method that does not need to design all combinations. Some parameter combination repeatability experiments are random repeatability experiments, mainly to verify the accuracy and repeatability of the experimental results.

(33)You have to write A (W) B (mm/s) C (g/min) H (mm) h (mm) W (mm)  instead of A/w B/mm/s C/g/min H/mm h/mm W/mm  

ResponseMany thanks for this suggestion.

Has been revised in the paper.In section 3.2

Table 4 has been modified to A (W) B (mm/s) C (g/min) H (mm) h (mm) W (mm)

(34)Edit (5), (6), (7), (8) and (9) equations like (1), (2), (3) or (4)

ResponseMany thanks for this suggestion.

Has been revised in the paper.

Please see attachment for revised manuscript

Reviewer 3 Report

Dear Authors,
Thank you for interesting paper regarding numerical simulation and parameter optimization of superalloy coating laser cladding.

After readinng of your manuscript, I have some suggestions and questions as below.

1. The mistakes or inadequate words (for the context od the sentence) can be found. Check the manuscript, please.

2. Experimental equipment - section 3.1. Chemical composition of Inconel 718 should be provided.

3. The scale of figure 1 is not visible. Please, correct it.

4. The values of laser power P in the caption of Figure 2 should be added.

5. Descriptions of Figure 3, 9 are not good visible. Please, correct it.

6. Figure 4 should be located below the text describing image. This figure is not well readable. If it is posible, please correct it.

7. The column markings in Table 4 (i.e. A,B,C) do not corresponds with the Table 3. The desciption column in Table 4 should be extened to full description or the same short cut for both table 3 and 4.

8. How the cross-sectional of Inconel 718 laser cladding was prepared? What device to structure observations was used (optical micrscope, scanning elentron microscope)? It is not described in the manuscript. Please, correct it.

9. The quality of figure 14 is very low. The ratio of brightness and contrast is incorrectly set.

Author Response

Dear Reviewers:

Thank you for your letter and for the reviewers' comments concerning our manuscript entitled "Numerical simulation and multi-objective parameter optimization of Inconel718 coating laser cladding" (ID#coatings-1712737). Those comments are valuable and helpful for revising and improving our paper and the critical guiding significance to our research. We have studied comments carefully and made the correction which we hope to meet the approval. Revised portions are marked in red in the paper. Point-by-point responses to the reviewers’ comments are as follows:

Reviewers' comments and responses:

  • The mistakes or inadequate words (for the context od the sentence) can be found. Check the manuscript, please.

ResponseMany thanks for this suggestion. 

The paper has been carefully checked and revised.

  • Experimental equipment - section 3.1. Chemical composition of Inconel 718 should be provided.

ResponseMany thanks for this suggestion. 

Has been revised in the paper.In section 3.1

Inconel 718 chemistry has been added to the paper. the chemical composition of Inconel718 powder is 0.08%C, 0.35%Si, 0.35%Mn, 55%Ni, 21%Cr, and the rest is iron

  • The scale of figure 1 is not visible. Please, correct it.

ResponseMany thanks for this suggestion. 

Image has been modified.In section 2.1.1

  • The values of laser power P in the caption of Figure 2 should be added.

ResponseMany thanks for this suggestion. 

Has been revised in the paper.In section 2.2.1

The value of laser power P in the title of Figure 2 has been increased.

  • Descriptions of Figure 3, 9 are not good visible. Please, correct it.

ResponseMany thanks for this suggestion. 

Figures 3 and 9 have been modified.

  • Figure 4 should be located below the text describing image. This figure is not well readable. If it is posible, please correct it.

ResponseMany thanks for this suggestion. 

Figure 4 has been located below the description in the text. The figure mainly illustrates the comparison of experimental and simulation results.

  • The column markings in Table 4 (i.e. A,B,C) do not corresponds with the Table 3. The desciption column in Table 4 should be extened to full description or the same short cut for both table 3 and 4.

ResponseMany thanks for this suggestion. 

Has been revised in the paper. In section 3.2

Table 4 has been modified to use the same full description as Table 3.

  • How the cross-sectional of Inconel 718 laser cladding was prepared? What device to structure observations was used (optical micrscope, scanning elentron microscope)? It is not described in the manuscript. Please, correct it.

ResponseMany thanks for this suggestion. 

Has been revised in the paper. In section 3.1

the section of Inconel 718 laser cladding is cut along the vertical direction of the cladding layer by wire cutting equipment, and then the section is ground and polished with sandpaper.Finally, the sample was corroded with 4% nitric acid alcohol for about 30s, and the surface of the sample was wiped with alcohol and dried.The cross-section of the cladding layer was observed using a super-depth-of-field microscope.

  • The quality of figure 14 is very low. The ratio of brightness and contrast is incorrectly set.

ResponseMany thanks for this suggestion. 

Image has been modified.In section 5

Please see attachment for revised manuscript

Round 2

Reviewer 2 Report

The paper discusses numerical simulation and offers an optimal combination of parameters.

In the abstract it does not refer to the experimental part.

All the theoretical introduction is good but it is not applied to the study. Also in the introduction it does not make reference to the experimental part and how it is going to be related to the theoretical study.

In equation (1) it says that r, C and l are a function of temperature and temperature varies with time so r, C and l should also vary with time.

In equations (1) and (2) the letter Q appears, but it means different things and can be misleading. The same happens with h in equations (3) and (4).

In point 2.1.2 it says "based on the previous experimental results..." In which ones?

The size of the figures of the legends and axes are very small.

ARTICLEINFO     insert space between “ARTICLE” and “INFO”

Multi-objective optimization     change by           “multi-objective optimization”

“element technology, the influence”    change “,” by “.”  “element technology. The influence”

“element technology, First of all”            change “,” by “.”  “element technology. First of all”

“temperature field,establishes”                              change by           “temperature field, establishes”

“thermal conductivity,  T is the”               change by           “thermal conductivity, T is the”

“distribution function,  a is the internal”             change by           “distribution function, a is the internal”

“heat source,  t is the heat”                       change by           “heat source, t is the heat”

“transfer time,  C,l   changes”                  change by           “transfer time, C, l   changes”

Check all blanks after each dot and after each comma

In (“) you say that “physical properties do not change with temperature” but after equation (1) you say that density (r) change with temperature.

If l and r depend on T then in equation (1) it should include l(T) and r(T) and as T varies with time then the equation would be

Table 1 Temperature in K.

Include a table with the composition of the materials 45 steel and Inconel718 alloy.

Include a table with the powder composition, mesh …

Why is important the shape and dimension of the base plate?

Increase the size of the micron bar of Fig.1

Figure 2a and figure 2b, to which laser energy does each image correspond? If is 1200W and 2400W, the maximum temperature 2234K and 3938K are different than in the images.

What is the accuracy of temperature measurements?, Degrees Celsius?, tenths of a degree?

In fig 2a and Fig2b you write 1325 (without decimals) or 18.982 (with 3 significant figures).

“When the laser power is P=2400W (Fig.2 (b)), the temperature in the molten pool is too high and exceeds the boiling point of the alloy material, the cladding Inconel718 alloy may be vaporized” What is the boiling point? This data do not appear in table 1.

It is impossible to read data in Fig.3

Fig.4 show different between simulations and experiment results. But in the text you do not talk about experimental test.

Fig.5 Velocity is written with a lowercase v.

Fig 5 you show the range of speed that can form a better molten pool morphology. Can you show anyone that form a poor molten pool?

It is impossible to read data in Fig.6

Fig.7 show different between simulations and experiment results. But in the text you do not talk about experimental test.

Table 2 Laser powder (W), scan speed (mm/s), width (mm), depth (mm)

Table 2 How many measurements have you made to determine the error in %?

Why the substrate size is 100mmx100mmx10mm and in the simulation the base plate (150×50×10 mm) is different?

You have write in the same way with mm at the end or with mm after each figure.

After Fig.2 you write “Therefore, the laser power range is selected as 1200W-2400W”

Fig.9 It is impossible to read the words inside the image.

Fig.9 You have to write 9.3kW instead of 9.3KW

It is necessary more data about the 3kW fiber laser cladding machine, powder feed, super depth of field microscope, Inconel718 alloy powder… (manufacture, brand, model, …)

Table 3 Laser power (W), Scanning speed (mm/s), powder feeding rate (g/min)

Table 4. If you write the table in this way:

A/w B/mm/s C/g/min H/mm h/mm W/mm h j

11 1200 15 18 0.416 0.466 1.635 0.472 3.93

17 1200 19 16 0.375 0.334 1.594 0.529 4.25

1 1200 19 20 0.235 0.278 1.412 0.458 6.01

10 1200 23 18 0.356 0.330 1.381 0.519 3.88

16 1800 15 16 0.665 0.432 2.134 0.606 3.21

12 1800 15 20 0.445 0.354 1.758 0.557 3.95

3 1800 19 18 0.488 0.418 1.786 0.539 3.66

5 1800 19 18 0.478 0.435 1.874 0.522 3.92

13 1800 19 18 0.495 0.445 1.747 0.526 3.53

14 1800 19 18 0.503 0.441 1.755 0.531 3.49

15 1800 19 18 0.482 0.407 1.865 0.542 3.87

9 1800 23 16 0.512 0.372 1.732 0.579 3.38

6 1800 23 20 0.335 0.325 1.554 0.507 4.64

2 2400 15 18 0.763 0.458 2.087 0.625 2.73

7 2400 19 16 0.792 0.514 2.574 0.606 3.26

8 2400 19 20 0.554 0.448 1.882 0.553 3.40

4 2400 23 18 0.623 0.475 1.925 0.567 3.09

Why weren't all possible combinations made?

Why were some combinations of parameters repeated and others not?

You have to write A (W) B (mm/s) C (g/min) H (mm) h (mm) W (mm) h   j  instead of

A/w B/mm/s C/g/min H/mm h/mm W/mm h j

Edit (5), (6), (7), (8) and (9) equations like (1), (2), (3) or (4)

Include DOI of publications.

Author Response

Thank you very much for the teacher's suggestion, which is very helpful for the improvement of my thesis, thank you again.

Please refer to the attachment for specific modification instructions.

Reviewer 3 Report

Thank you very much for the corrections. I accept the manuscript in this form to publication.

Author Response

Thank you very much for the teacher's suggestion, which is very helpful for the improvement of my thesis, thank you again.